# Effect and Correlation of *Rosa roxburghii* Tratt Fruit Vinegar on Obesity, Dyslipidemia and Intestinal Microbiota Disorder in High-Fat Diet Mice

**DOI:** 10.3390/foods11244108

**Published:** 2022-12-19

**Authors:** Jiuchang Li, Jun Zhang, Yulong Zhang, Yuanyuan Shi, Dandan Feng, Yunyang Zuo, Ping Hu

**Affiliations:** School of Liquor and Food Engineering, Guizhou University, Guiyang 550025, China

**Keywords:** *Rosa roxburghii* Tratt fruit vinegar, obesity, dyslipidemia, gut microbiota regulation, correlation

## Abstract

To investigate the effect of *Rosa roxburghii* Tratt fruit vinegar (RFV) on the intervention of obesity and hyperlipidemia and its potential mechanism, a high-fat diet (HFD)-induced obesity model in mice was established and gavaged with RFV, saline and xuezhikang for 30 consecutive days, respectively. The results showed that RFV supplementation significantly reduced fat accumulation, and improved dyslipidemia and liver inflammation in HFD mice. RFV intervention for 30 days significantly improved the diversity of gut microbiota and altered the structure of gut microbiota in HFD mice. Compared with the model group (MC), the ratio of *Firmicutes* to *Bacteroidetes* at least decreased by 15.75% after RFV treatment, and increased the relative abundance of beneficial bacteria (*Proteobacteria*, *Bacteroidetes*, *Lactobacillaceae*, *Bacteroides*, *Akkermansia*,) and decreased the relative abundance of harmful bacteria (*Ruminococcaceae*, *Erysipelotrichaceae*, *Ruminococcaceae _UCG-013*, *Lachnospiraceae*, *Allobaculum*, *Actinobacteria*). Spearman’s correlation analysis revealed that *Erysipelotrichaceae*, *Allobaculum*, *Lachnospiraceae*, *Ruminococcaceae*, *Ruminococcaceae_UCG-013*, *uncultured_bacterium_f_Lachnospiraceae and Desulfobacterota* were positively correlated (*p* < 0.05) with the body weight of mice, while *Proteobacteria* was negatively correlated (*p* < 0.05) with the body weight of mice. The two main bacteria that could promote dyslipidemia in obese mice were *Actinobacteria* and *Firmicutes*, while those that played a mitigating role were mainly *Bacteroidetes.* It is concluded that RFV plays an important role in the intervention of obesity and related complications in HFD mice by regulating their gut microbiota.

## 1. Introduction

*Rosa roxburghii* Tratt (RRT) usually grows in mountainous areas between 500–2500 m above sea level and is widely cultivated in Guizhou Province, southwest of China. RRT fruit is a nutritious fruit that is rich in vitamin C (VC) (1300–3500 mg/100 g), superoxide dismutase (SOD) (4609.26–5797.48 U/g), phenolic compounds (108.9–156.3 mg GAE per g dry weight), flavone (66.23–114.2 mg RE per g dry weight) and other pharmacological active ingredients [1,2,3]. RRT fruit has the highest vitamin C in all fruit known and is known as the “king of vitamin C” [1]. Some studies have shown that RRT fruit and its extract have various functional properties, such as antioxidant activity [4], anti-inflammatory activity, hypoglycemic activity, lipid-lowering activity and regulation of intestinal microbiota disorder [5,6,7] and other functions. However, RRT is rich in tannins and has a strong bitter and astringent taste, which makes it difficult for many consumers to accept. The tannin content in RRT juice can be reduced or masked by adding gelatin, chitosan or bentonite to clarify, or adding sweeteners such as honey to improve the taste [8]. However, the effects of these treatments are poor, and the loss of VC and SOD in RRT during storage is serious. Numerous studies have shown that probiotic fermentation can effectively remove or reduce the anti-nutritional factors such as tannins and phytic acid in raw materials and can produce food with higher nutritional value [9,10,11]. Fermentation provides a natural solution to the astringency of these fruits and vegetables.

Using probiotics to ferment fruits and vegetables can not only remove the bitter taste and astringency in raw materials, but also produce bioactive metabolites, resulting in unique taste and more balanced nutrition [12,13,14] Vinegar is a fermented seasoning produced by alcohol fermentation and acetic acid fermentation. Studies have shown that vinegar has the potential to improve hypertension, hyperlipidemia, obesity, diabetes and regulate intestinal flora [15,16,17]. Mohamad et al. reported that the gavage of obese mice using coconut water vinegar augmented the abundance of *Akkermansia* and *Bacteroides* [18]. Similarly, Hosoda et al. found that ginkgo vinegar inhibited HFD-induced weight gain in mice and reduced the size of adipocytes [19].

Obesity seriously affects people’s physical health and quality of life, and has become a public health problem of global concern. Many studies have shown that obesity is associated with many chronic diseases such as atherosclerosis, cardiovascular disease, inflammatory bowel disease, hyperlipidemia, and non-alcoholic fatty liver disease [8,20,21]. Obesity caused by HFD disrupts the balance of gut microbiota and triggers syndromes such as dyslipidemia, steatosis, and liver inflammation [22]. More and more studies have confirmed the causal relationship between obesity and intestinal microbiota imbalance [22,23,24]. Therefore, regulating gut microbiota balance to combat obesity and related complications is a new potential treatment.

Our previous study found that in the production process of RFV, the mixed fermentation of lactic acid bacteria and RRT resulted in low tannin content, good sensory quality and stable nutrient composition. However, the functional properties of RFV were not well understood. The purpose of this study was to evaluate the anti-obesity potential of RFV in an HFD-induced obesity mouse model, and to explore the effect of RFV on the improvement of dyslipidemia and regulation of intestinal microbiota in obese mice, and clarify the correlation between RFV regulation of intestinal microbiota and the improvement of obesity and dyslipidemia in mice.

## 2. Materials and Methods

### 2.1. Dietary Supplementation and RFV Preparation

Normal diet (ND) for experimental mice was purchased from Tianqin Biotechnology Co., Ltd. (Hunan, China), HFD (78.8% normal diet, 10% egg yolk powder, 10% fat, 1% cholesterol, 0.2% salt of cholate). The ND was kept at room temperature, but the HFD was kept at 4 °C. The harvested RRT fresh fruit were selected, cleaned, beaten, filtered, ultra-high temperature instantaneous sterilization (UHT), and aseptically filled. Then, an appropriate amount of RRT juice was taken to dilute 5 times (DRRTJ) with sterile water and set aside. Production of RFV consisted of two fermentation stages: alcoholic fermentation (7 days) and acetic fermentation (8 days). A flowchart of the process is presented in Figure 1. Three lactobacilli (homemade) used were Lactobacillus rhamnosus H3 (CCTCC NO: M2016525), Lactobacillus paracaseiSR10-1 (CCTCC NO: M2016527), and Lactobacillus fermentum GZSC-1 (CCTCC NO: M2017847), using a ratio of 1:1:1. The fruit wine yeast was purchased from Angel Yeast Co., Ltd. (Hubei, China). Acetobacter (CGMCC1.41) was purchased from China General Microorganism Culture Collection and Management Center.

### 2.2. Measurement of Chemical Functional Components

Total phenolic content was measured by Folin–Ciocalteu reagent using gallic acid as a standard [7]. Total flavonoid content was measured by NaNO2-Al(NO3)3-NaOH reagent using rutin as a standard [7]. The measurement of total acid was performed by acid-base titration method [12]. The content of VC was measured by colorimetric method [12]. Measurement of SOD content was performed via SOD kit (SOD kit purchased from Nanjing Jiancheng Institute of Biological Engineering, Jiangsu, China) according to the kit instructions.

HPLC was used to analyze organic acids in RFV based on the literature [25] with slight modifications. Separation was conducted using an Agilent Z0RBAX SB-AQ column (250 mm × 4.6 mm, 5 µm, American Agilent Corporation (Santa Clara, CA, USA) with an Agilent1260 VWD detector. The relevant parameters of the mobile phase are as follows: monopotassium phosphate buffer solution (0.02 mol/L, pH 2) the ratio of methanol was 95:5, the injection volume was 10 µL and the flow rate was 0.8 mL/min, the spectra were recorded at 210 nm, the column temperature was 35 °C. The experiment was conducted three times for each sample.

### 2.3. Animal Experiment Design

All the animal experiments were allowed and performed according to the “Administrative Regulations on Laboratory Animals”, and were reviewed and approved by the Laboratory Animal Ethics Sub-Committee of Guizhou University (No. EAE-GZU-2021-E002). All 48 of 4-week-old Kunming male mice were purchased from Tianqin Biotechnology Co., Ltd. (Hunan, China), weighing approximately 22 ± 3 g, and kept under specific pathogen-free (SPF) conditions (23 ± 2 °C at 50 ± 5% relative humidity) with a 12 h light/dark cycle, and water and food were consumed ad libitum. After one week of acclimatization, mice were randomly divided into 6 groups (n = 8 per group) according to their body weight according to grouping number of mice a study reported [18]. The grouping was as follows: (1) normal group (NC): normal feed, 0.9% saline; (2) model group (MC): HFD, 0.9% saline; (3) positive group (PC): HFD, 0.5 mg/mL xuezhikang capsules (The main ingredient is Monascus extract); (4) RFV low dose group (LD): HFD, 10% RFV; (5) RFV medium dose group (MD). HFD, 30% of RFV; (6) RFV high dose group (HD): HFD, 50% of RFV. The gavage dose of an individual mouse was 1% of its body weight (0.1 mL/10 g), and the maximum gavage volume did not exceed 0.4 mL. The gavage dose was performed according to previous reported [12]. The mice were gavaged daily and continuously for 30 days. Body weight was measured once a week during the rearing period. At the end of day 30, mice were fasted overnight and sacrificed to obtain organs, tissues, and serum samples, and they were stored at −80 °C until use.

### 2.4. Organ Index of Mice Were Measured

The organ index is mainly used as an index to assess whether the feeding of HFD causes the hypertrophies of liver, spleen, kidney, and heart of mice. After the dissection, the liver, spleen, kidney and heart were sequentially collected, rinsed with normal saline, blotted with filter paper, and weighed. The organ indices were calculated according to the following formula:Organ index (%) = (mouse organ weight/mouse body weight) × 100%(1)

### 2.5. Serum and Liver Biochemical Analysis

The levels of total cholesterol (TC), triacylglycerol (TG), high-density lipoprotein cholesterol (HDL-C), low-density lipoprotein cholesterol (LDL-C) and total superoxide dismutase (SOD) in serum were measured using commercial kits (Jiancheng Bioengineering Institute, Nanjing, China). Hepatic malondialdehyde (MDA), Glutathione (GSH), catalase (CAT) were measured using commercial kits (Beijing Solarbio Technology Co. Ltd., Beijing, China).

### 2.6. Histological Analysis

Mouse livers were fixed in 4% neutral formaldehyde solution for 24 h. The samples were processed using a series of fractionated ethanol solutions to dehydrate, embedded in paraffin, and sectioned into 5 μm thick sections via Leica CM 1900-1-1 freezing microtome (Leica Instrument Co., Ltd, Shanghai, China). All tissues were stained with hematoxylin and eosin (H&E), and pathological changes were observed using a light microscope (Nikon Eclipse E100, Japan Nikon).

### 2.7. Gut Microbiota Analysis

After dissection of the mice, the cecum was collected and transferred into sterile EP tubes to be immediately frozen using dry ice, and stored at −80 °C for microbiota analysis. Total DNA was extracted from the fecal samples using a Biomarker Soil Genomic DNA Kit (Biomarker Technologies, Beijing, China) according to the manufacturer’s instructions. The quality of the extracted DNA was assessed by NanoDrop 2000 UV microspectrophotometer and 1.8% agarose electrophoresis (120 V, 40–45 min). The extracted DNA was amplified with barcoded primers of variable 3–4 region of the 16S RNA gene with the primers 338F and 806R (fwd 5′-ACTCCTACGGGAGGCAGCA-3′ and rev 5′-GGACTACHVGGGTWTCTAAT-3′). All the samples were sequenced and bioinformatics analyzed on the PacBio sequencing platform by Beijing Biomarker Technologies Co., Ltd. The community diversity and community richness were assessed with alpha diversity including Shannon, and Chao 1 indexes.

### 2.8. Statistical Analysis

Data are presented as mean ± standard deviation (SD). The data were analyzed by one-way analysis of variance (ANOVA). The *p* < 0.05 (*) indicates a significant difference, and *p* < 0.01 (**) indicates a highly significant difference.

## 3. Results

### 3.1. Bioactive Compounds in RFV

As shown in Table 1, SOD, VC, total phenols, total flavonoids and malic acid were significantly reduced after fermentation of DRRTJ. The contents of acetic acid, lactic acid and other active components in RFV were significantly increased (*p* < 0.05), and the total organic acid content was 3.355 times higher than that before fermentation. The contents of total phenols (251.32 ± 1.82 mg/100 g), total flavonoids (98.89 ± 1.88 mg/100 g) in RFV and types of organic acids were higher than those in the apple vinegar (No.14) reported in the work of [25]. The results fully indicated that the nutrient composition of RFV was more balanced after probiotic fermentation. Numerous studies have shown that foods rich in active substances such as VC, SOD, polyphenols, flavonoids, acetic acid, organic acids have antioxidant effects and alleviate chronic diseases such as obesity [1,12,26]. Therefore, the rich nutrients in RFV lay a material foundation for further exploration of alleviating obesity and dyslipidemia.

### 3.2. Anti-Obesity Potential of RFV

As shown in Figure 2, HFD feeding significantly increased the body weight of mice by 17.813% compared to the ND group (*p* < 0.05), indicating that long-term feeding HFD caused a significant increase in body weight of mice. However, RFV supplementation significantly inhibited body weight gain in mice, and there was no significant difference in body weight when compared to the NC group after 30 d of a continuous gavage of RFV (*p* > 0.05), indicating that RFV effectively inhibited body weight gain in HFD-induced mice. In general, weight gain is usually accompanied by visceral fat accumulation. The mice in MC group had the largest liver index (4.93 ± 0.17), while the maximum value of other organ indices were not observed in the MC group (Table 2), which may be related to the fact that obesity easily causes fat accumulation in the liver. The value of liver indexes of other groups were lower than that of NC group, indicating the beneficial effect on reducing liver fat accumulation through experimental treatment.

### 3.3. Potential Preventing Dyslipidemia of RFV

Atherosclerosis, cardiovascular disease, and metabolic syndrome are considered to be closely associated with LDL-C and HDL-C abnormalities [21,22,23]. Therefore, the measurement of LDL-C and HDL-C indicators is informative for the assessment of obesity, hyperlipidemia and hypertension. As shown in Figure 3, compared with the NC group, HFD intervention caused a significant increase in serum lipid biochemical parameters, including TC, TG, and LDL-C levels (*p* < 0.01). In contrast, the HDL-C level was significantly decreased (*p* < 0.01).There results indicated that HFD caused dyslipidemia in obese mice. Compared with MC group, TC, TG, and LDL-C levels were significantly decreased (*p* < 0.01), and HDL-C levels were significantly increased in the PC, LD, MD, and HD groups. TC, TG, and LDL-C indicators were down-regulated by at least 16.16%, 17.97%, and 20.91%, respectively, and HDL-C indicators were up-regulated by 44.53% after RFV supplementation. There was no significant difference in TG and HDL-C levels in the HD group compared with the NC group, indicating that RFV supplementation had a significant interventional effect in regulating serum abnormalities in obese mice (*p* < 0.05).

### 3.4. Hepatic Antioxidant Power of RFV

Figure 4 shows hepatic antioxidant power of RFV. Compared to the NC group, HFD induction caused a decrease in CAT, GSH, and SOD, and an increase in MDA in mice livers. Compared with the MC group, CAT, GSH, MDA, and SOD were significantly improved in each group by gavage of RFV and xuezhikang capsule. CAT, GSH, and SOD indexes were increased by at least 7.60%, 17.33%, and 29.38%, and MDA was decreased by at least 29.78% after RFV treatment, and the improvement of the SOD index was better than that of the PC group (27.28%) (Figure 4D). CAT, GSH, MDA and SOD indexes were effectively improved after RFV supplementation, which indicated that RFV had a positive effect on improving the oxidative stress ability of the liver in HFD mice.

### 3.5. Histological Trait of Mice Liver

HFD can cause liver inflammation, injury and steatosis, which can lead to non-alcoholic steatohepatitis and non-alcoholic fatty liver disease [24]. Histopathological changes in the liver are shown in Figure 5. The liver tissue structure in the NC group was intact and clear, with a standard lobular structure and an orderly arrangement of hepatocyte distribution. Some nuclei in the MC group were extruded into the cell membrane, and hepatocyte balloon degeneration, lipid droplet accumulation, and inflammation were observed. Compared with the MC group, hepatocyte lipid droplets were slightly reduced in the LD and MD groups, a small number of hepatocytes showed vacuolated structures, and the steatosis was not completely improved. Nevertheless, the decrease of lipid droplets in HD and PC groups was greater, and the structure of liver tissue was more similar to that of NC group. Histological straits showed that the appropriate amount of RFV ameliorated hepatic steatosis induced by HFD in mice.

### 3.6. Effects of RFV on Gut Microbiota in Obese Mice

A dilution curve was used to assess whether sequencing depth reflects the microbial diversity and richness in a sample. As shown in Figure 6A, the dilution curve of the samples reached the plateau phase when the sequence number exceeded 6000, indicating that most of the bacteria in the sample were identified. Shannon and Chao 1 indices are often used to measure samples’ microbial diversity and richness. The greater the Shannon and Chao 1 index, the greater the community diversity and richness of the sample [23]. Figure 6B,C showed that the α-diversity (the Chao 1 and Shannon’s diversity parameter) of mice was significantly decreased in the HFD-induced mice group compared to the NC group, suggesting that the HFD decreased the diversity of gut microbiota in mice. Species richness and diversity were not effectively improved in the LD and MD groups, and even tended to decline. Nevertheless, high doses of RFV significantly enhanced the diversity and richness of intestinal microbiota in HFD-induced obese mice, and even reached and exceeded the effect of the PC group.

As shown in Figure 7A, the *Firmicutes*, *Proteobacteria*, and *Bacteroidetes* were the dominant phylum species common to all groups, with a relative proportion of more than 90%, which is consistent with existing studies [27,28,29]. Compared with the NC group, the relative abundance of *Firmicutes* in the MC group sharply increased and became the absolute dominant microbial population (*p* < 0.01), while the relative abundance of *Bacteroidetes* was significantly decreased (*p* < 0.01), making the ratio of *Firmicutes* to *Bacteroidetes* (F/B) severely imbalanced. *Firmicutes* have been shown to promote energy absorption and weight gain, resulting in higher F/B ratios in obese mice [30]. However, RFV reversed this situation and significantly decreased the (F/B) ratio, in which the HD group had the most significant effect, with a 66.08% reduction in the F/B ratio (shown in Figure 7B,C). Feeding food containing active compounds such as polyphenols, flavonoids, and organic acids can lead to lower F/B ratio and body weight in HFD-induced mice [31,32]. The results of this study also support this idea.

The top 10 groups in terms of relative abundance at the family level were shown in Figure 8A. *Ruminococcaceae* and *Erysipelotrichaceae* were considerably increased (*p* < 0.01) in MC group compared to NC group, while *Muribaculaceae*, *Bacteroidaceae*, *Lachnospiraceae*, and other families were dramatically decreased (*p* < 0.01). In obese mice, however, gavage RFV changed the abundance and structure of gut microbiota. The relative abundance of *Bacteroidaceae* and *Lactobacillaceae* was significantly increased (*p* < 0.01), and the relative abundance of *Ruminococcaceae* and *Erysipelotrichaceae* was significantly decreased after gavage of RFV compared to the MC group. However, the relative abundance of *Ruminococcaceae* was still increasing in the PC group, and the relative abundance of *Erysipelotrichaceae*, *Bacteroidaceae*, and *Lactobacillaceae* was not effectively improved. This indicated that the chemical drugs had no regulatory effect on the intestinal flora of obese mice. Comparative analysis showed that RFV regulated the intestinal microbiota balance in obese mice, and had a regulatory effect on chronic diseases caused by intestinal microbiota imbalance.

Species with relative abundances higher than 1% at the genus level are shown in Figure 8B. The groups differed in both structure and abundance. The abundance of *Ruminococcaceae-UCG-013*, *Allobaculum*, *Ruminococcaceae-UCG-014*, uncultured*-bacterium-f-Lachnospiraceae* and *Faecalibaculum* in the MC group was significantly higher than that in the NC group (*p* < 0.05). Nevertheless, the relative abundance of uncultured*-bacterium-f-Muribaculaceae*, *Lachnospiracea-NK4A136*, *Bacteroides*, uncultured*-bacterium-f-Desulfovibrionaceae*, *Parabacteroides* and *Alloprevotella* were significantly reduced (*p* < 0.05). This result suggests that HFD induction causes gut microbial dysbiosis at the genus level in mice as expected. However, *Enterococcus*, uncultured*-bacterium-f-Muribaculaceae*, *Lactobacillus*, *Alloprevotella*, *Bacteroides*, uncultured*-bacterium-f-Desulfovibrionaceae* and *Parabacteroides* increased significantly after gavage of RFV. *Lactobacillus* was significantly higher in LD and MD groups than in NC and MC groups (*p* < 0.01), and *Bacteroides* was significantly higher in all three dose RFV groups than in theMC group. The relative abundance of *Akkermansia* in HD group was 9.40%, which was significantly higher than that in the MC group (3.91%). *Akkermansia* was considered to be a beneficial bacterium with anti-inflammatory effects and other effects [32]. However, the relative abundance of *Ruminococcaceae-UCG-013*, *Lactobacillus*, *Bacteroides* and *Akkermansia* was not effectively regulated in the positive control group. From the above analysis, the regulation effect of high-dose RFV on intestinal microbiota structure and relative abundance at the phylum, family, and genus level was significantly better than that of the positive control group, which indicated that gavage of high-dose RFV had a significant effect on regulating intestinal microbiota disorder caused by obesity.

### 3.7. Correlation between Obesity-Related Parameters and Microbiota

The 10 bacteria with the highest relative abundance at the level of each phylum, family and genus were selected for correlation analysis with obesity-related indicators. Correlation analysis showed that 11 key OTUs were positively or negatively correlated with parameters related to metabolic disorders in obesity, including weight body gain, TC, TG, LDL-C, HDL-C, CAT, SOD, GSH and MDA (Figure 9). Bacteria that were significantly positively correlated with the weight gain of mice (*p* < 0.05) included *Allobaculum*, *Erysipelotrichaceae*, uncultured*_bacterium_f_Lachnospiraceae*, *Ruminococcaceae_UCG-013*, *Ruminococcaceae*, *Desulfobacterota*, *Lachnospiraceae*, *Firmicutes* and *Actinobacteria.* However, *Proteobacteria* was significantly negatively correlated with the weight gain of mice (*p* < 0.05). It is worth noting that *Proteobacteria* was inconsistent with our results in a similar study by Sheng [33] et al., but it was consistent in a similar study by Jiao [34] et al., and this difference may be related to the polyphenol content in the substance supplemented to HFD mice. The bacteria that can promote dyslipidemia in obese mice are mainly *Actinobacteria* and *Firmicutes*, while those that play a mitigating role are mainly *Bacteroidetes*. The main five species of bacteria that played an effect on oxidative stress in mice were *Actinobacteria*, *Firmicutes*, *Lachnospiraceae*, *Desulfobacterota*, and *Bacteroidetes*, among which *Actinobacteria*, *Firmicutes*, and *Bacteroidetes* had the most significant effect on the level of SOD. In summary, these bacteria were significantly associated with weight gain, dyslipidemia, and oxidative stress in mice, indicating these bacteria might be the most efficient genera contributing to preventing the development of obesity. It can be seen from the analysis in Section 3.6 that supplementation of RFV promoted the enrichment of beneficial bacteria *Bacteroidetes* and *Proteobacteria* while reducing the relative abundance of *Lachnospiraceae*, *Desulfobacterota*, *Ruminococcaceae*, *Ruminococcaceae_UCG-013*, uncultured*_bacterium_f_Lachnospiraceae*, *Erysipelotrichaceae*, *Allobaculum*, *Actinobacteria* and *Firmicutes*. These analyses suggest that RFV may alleviate obesity and its associated chronic diseases in HFD-induced obesity mice by regulating the relative equilibrium of their gut microbiota.

## 4. Discussion

Numerous studies have shown that gut microbiota and its metabolites are inextricably linked to the development of obesity and type 2 diabetes [35,36]. Foods rich in bioactive components such as polyphenols, organic acids and terpenoids have an important impact on intestinal microorganisms and may alleviate obesity and related metabolic syndrome [7,37,38]. Acetic acid helps reduce fat accumulation and plays an important role in the fight against obesity [15,26]. The RFV used in this study was rich in various bioactive substances such as acetic acid (35.522 ± 1.125 mg/mL) and polyphenols (251.32 ± 1.82 mg/100 g) (Table 1). From the above analysis, RFV has a potential role in improving intestinal flora dysbiosis and alleviating metabolic syndrome such as obesity. Therefore, it is necessary to explore the intervention effect of RFV on obesity-induced dyslipidemia, hepatic steatosis and intestinal microbiota by establishing a mouse obesity model.

In vivo evaluation results showed that HFD caused fat accumulation, dyslipidemia, liver damage and dysbiosis of intestinal flora in mice when compared to the NC group. However, the weight of HFD mice after RFV supplementation was not significantly different from that of the NC group (*p* > 0.05), showing that the weight gain of mice was effectively controlled (Figure 2). HFD significantly increased the levels of TC, TG and LDL-C in serum, and significantly decreased the levels of HDL-C in serum of mice, which was consistent with previous studies [27]. The levels of TG, LDL-C, HDL-C and TC in serum of obese mice were effectively improved after RFV intervention (Figure 3). Studies have shown that supplementation with RRT juice improves dyslipidemia in obese mice [1,7]. However, this study showed that RFV supplementation not only effectively improves the dyslipidemia in obese mice caused by HFD, but also the palatability of RRT juice based on the significant intervention effects of RFV on body weight, LDL-C, TC, TG and HDL-C levels in mice. Meanwhile, the results of SOD, MAD, CAT and GSH in mice liver showed that RFV supplementation improved the abnormal phenomena of SOD, MAD, CAT and GSH, and enhanced the ability of oxidative stress in HFD-induced obese mice (Figure 4).

H&E staining showed that the structure distribution and arrangement of hepatocytes in HD group were more similar to those in the NC group (Figure 5), indicating that a higher dose of RFV could help alleviate obesity and hepatocyte steatosis and other symptoms. A similar study by Mohamad et al. found that pineapple vinegar reduced body weight and alleviated associated inflammation in HFD rats [39]. Similarly, Beh et al. found synthetic vinegar to be equally effective in promoting weight loss and alleviating dyslipidemia [15]. Acetic acid can prevent the associated metabolic syndrome by reducing body weight in obese mice [26], and vinegar can reduce body weight, body fat mass and serum triglycerides in obese subjects [40]. However, further analysis of the underlying mechanisms by which they interfere with obesity and other related symptoms is still necessary. In this study, we hypothesized that the anti-obesity effect of RFV, improvement of dyslipidemia and hepatic oxidative stress level were closely related to gut microbiota. Therefore, 16S rRNA sequencing was performed to analyze the intestinal microbiota of mice in each group. The results of high-throughput sequencing showed that RFV intervention effectively improved intestinal microbiota dysbiosis in obese mice, and increased the diversity and abundance of beneficial microbiota (Figure 7, Figure 8 and Figure 9).

It was found that at the phylum level, the relative abundance of *Firmicutes* in the intestinal flora of HFD-induced obese mice were significantly increased, and the relative abundance of *Bacteroidetes* was significantly decreased, leading to a significant increase in the F/B ratio. It has been suggested that higher F/B ratio may be closely associated with obesity-related metabolic disorders and can be recovered by weight loss [41]. Studies have shown that *Firmicutes* have a higher efficiency of sugar metabolism than the *Bacteroidetes*, which favors energy absorption and promotes obesity [42]. A higher F/B ratio implies more energy intake and the possibility of being overweight [43]. The relationship between an increase or decrease in the F/B ratio and obesity is subject-dependent, and one study showed that: treatment of obese mice with Eurotium cristatum decreased the F/B ratio; similarly, treatment of obese dogs with Eurotium cristatum increased the F/B ratio [44]. This study showed that RFV intervention significantly altered the relative abundance of *Firmicutes* and *Bacteroidetes* and reduced the value of F/B, indicating that RFV had a functional role in regulating the gut microbiota to a normal level.

At the family level, HFD induction significantly increased the relative abundance of *Erysipelotrichaceae* and *Ruminococcaceae* and decreased the relative abundance of beneficial bacteria such as *Bacteroidaceae* and *Muribaculaceae* in the gut of HFD mice (Figure 8). There was a significantly positive correlation between *Erysipelotrichaceae* and complex carbohydrate intake [45]. *Erysipelotrichaceae* can enhance the host’s energy acquisition from the diet, thereby altering the host’s lipid distribution [46]. In addition, the abundance of *Erysipelotrichaceae* was found to be significantly increased in patients with inflammatory bowel disease [47], colorectal cancer [48], and other inflammatory diseases. The relative abundance of *Ruminococcaceae* is closely related to HFD [49,50]. However, the present study indicated that RFV intervention significantly reduced the relative abundance of *Erysipelotrichaceae* and *Ruminococcaceae*. Moreover, the *Muribaculaceae* were significantly increased in the HD group compared to the MC group. Studies have reported that *Muribaculaceae* can degrade food and polysaccharides to produce short-chain fatty acids (SCFAs), including succinic acid and propionic acid, and others [32]. SCFAs have been shown to have important anti-inflammatory properties and to balance glycolipid homeostasis [51]. A high dose of RFV could promote the proliferation of *Muribaculacea* in mice gut to produce SCFAs. RFV treatment reversed the dramatic decrease in *Bacteroidaceae*, reaching the level of the NC group in the LD and HD groups. A large number of studies showed that *Bacteroidaceae* was negatively correlated with obesity, which was helpful in reducing obesity [24,52,53,54]. Microbial populations and abundance significantly differed in the genus level. *Ruminococcaceae-UCG-013* and *Ruminococcaceae_UCG-014* are considered beneficial bacteria in the gut of type 2 diabetic rats [55,56]. The difference was that *Ruminococcaceae-UCG-013* and *Ruminococcaceae_UCG-014* were highly enriched in the MC group, while they were significantly decreased after RFV treatment (*p* < 0.01) (Figure 9). It was speculated that *Ruminococcaceae-UCG-013* and *Ruminococcaceae_UCG-014* might be positively correlated with obesity. *Akkermansia* belongs to the *Verrucomicrobia* family [57] and has been shown to reduce systemic disease LPS levels, weight gain, and fat mass in HFD-fed rats [58]. Numerous studies have shown that dietary polyphenols or flavonoids can promote the enrichment of *Akkermansia* in the intestines of mice [32,43]. In this study, the abundance of *Akkermansia* in the HD group was 58.4% higher than that in the MC group (0.0391) (Figure 8B), indicating that high-dose RFV contributed to the enrichment of *Akkermansia* in the intestine of mice. *Bacteroides* were largely absent after HFD induction, but started to be enriched again after gavage of RFV and had reached the level of the NC group in the LD and HD groups (Figure 8). Studies have shown that *Bacteroides* have the potential to prevent inflammatory diseases of the intestinal tract [58,59]. *Lactobacillus* can metabolize sugars to produce lactic acid and is often considered a probiotic [53], and its abundance was negatively correlated with fat and body weight in high-fat induced obesity rats [53,60]. This study showed that a significant increase in *Lactobacillus* was observed in the LD and MD groups compared with the MC group, indicating that *Lactobacillus* had an important role in reducing obesity.

In addition, our study showed that RFV did effectively alleviate the imbalance of intestinal microbiota in obese mice. Through correlation analysis, the relative balance of *Allobaculum*, *Erysipelotrichaceae*, *Lachnospiraceae*, *Desulfobacterota*, *Ruminococcaceae*, *Ruminococcaceae_UCG-013*, *Actinobacteria*, *Bacteroidetes*, *Firmicutes*, *Proteobacteria* had a significant impact on the weight gain and related chronic diseases of mice. Our findings provide a reference for further exploring the direct relationship between gut microbiota and obesity and other metabolic syndromes.

## 5. Conclusions

In order to clarify the ameliorating effect of RFV on dyslipidemia and other phenomena in obese mice, RFV was used to intervene in HFD-induced mice. It has been found that supplementation of HFD mice with different doses of RFV can improve body weight gain, dyslipidemia and intestinal flora dysbiosis caused by HFD. This study further revealed that high-dose RFV was more effective in controlling body weight gain, liver lipid droplet aggregation and regulating intestinal flora abnormalities. More importantly, the relative abundance and structure of the associated bacteria at the phylum, family and genus levels were closer to those of the NC group after gavage of high-dose RFV to HFD mice. These results suggest that the improvement of weight gain, dyslipidemia, and other indicators after supplementation by high-dose RFV are highly correlated with the genus-level *Allobaculum*, *Ruminococcaceae_UCG-013*, uncultured_bacterium_f_*Lachnospiraceae*, uncultured-bacterium-f-*Muribaculaceae*, *Bacteroides*, uncultured_bacterium_f*_Desulfovibrionaceae*, *Parasutterella*, *Akkermansia*, *Alloprevotella*, which proves that RFV has a large potential, as a functional food to prevent obesity and related chronic diseases.

## Figures and Tables

**Figure 1 foods-11-04108-f001:**
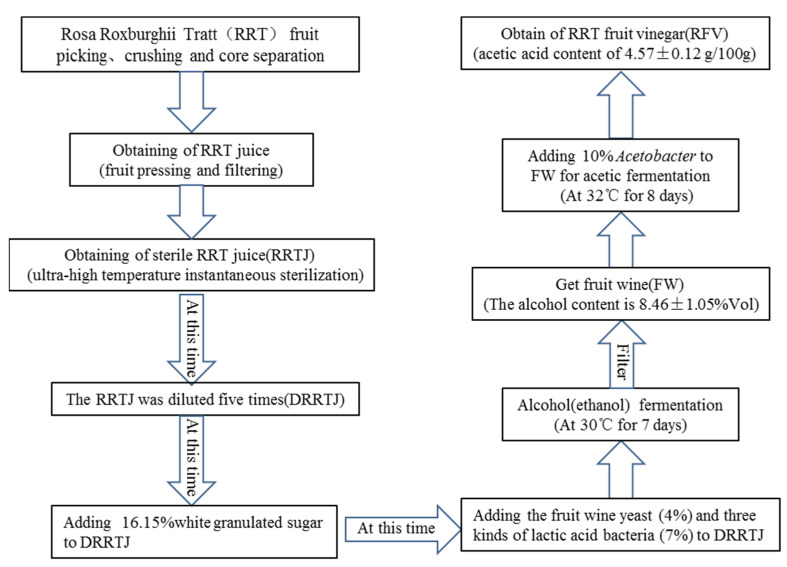
Production process of *Rosa roxburghii* Tratt fruit vinegar. All starter concentrations reached 10^8^ log CFU mL^−1^ in juices. Values are displayed as a bar chart plot with means (expressed as ‘+’), n = 3.

**Figure 2 foods-11-04108-f002:**
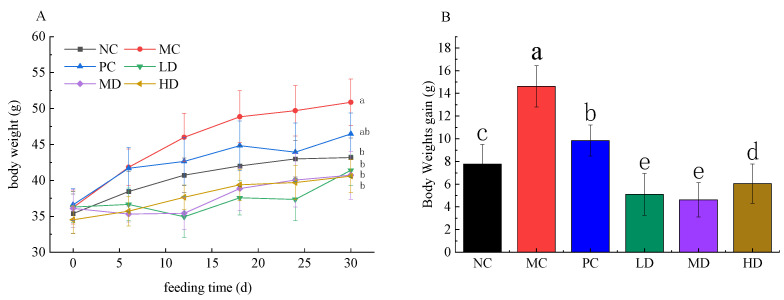
Body weight changes, body weight gain in mice fed experimental diets for 30 days. (**A**) Body weight changes; (**B**) body weight gain. Values are displayed as a bar chart plot with means (expressed as ‘+’), n = 3. a–e: bars with different letters are significantly different at *p* < 0.05 by Tukey’s post hoc test. NC: normal group (normal feed, 0.9% saline); MC: model group (HFD, 0.9% saline); PC: positive group (HFD, 0.5 mg/mL xuezhikang capsules); LD: RFV low dose group (HFD, 10% RFV); MD: RFV medium dose group (HFD, 30% of RFV); HD: RFV high dose group (HFD, 50% of RFV).

**Figure 3 foods-11-04108-f003:**
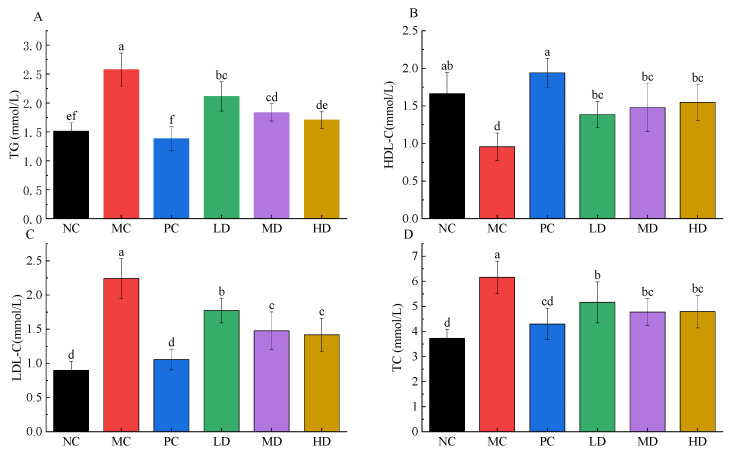
Lipid profiles in mice fed experimental diets for 30 days. Serum levels of TG (**A**); HDL-C (**B**); LDL-C (**C**) and TC (**D**) were measured in the experimental mice. Values are displayed as a bar chart plot with means (expressed as ‘+’), n = 3. a–f: bars with different letters are significantly different at *p* < 0.05 by Tukey’s post hoc test. NC: normal group (normal feed, 0.9% saline); MC: model group (HFD, 0.9% saline); PC: positive group (HFD, 0.5 mg/mL xuezhikang capsules); LD: RFV low dose group (HFD, 10% RFV); MD: RFV medium dose group (HFD, 30% of RFV); HD: RFV high dose group (HFD, 50% of RFV).

**Figure 4 foods-11-04108-f004:**
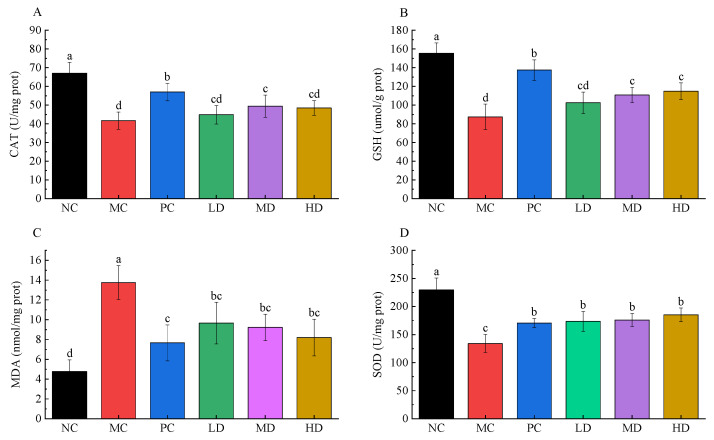
Liver oxidative stress in mice fed experimental diets for 30 days. (**A**) CAT; (**B**) GSH; (**C**) MDA; (**D**) SOD. Values are displayed as a bar chart plot with means (expressed as ‘+’), n = 3. a–d: bars with different letters are significantly different at *p* < 0.05 by Tukey’s post hoc test. NC: normal group (normal feed, 0.9% saline); MC: model group (HFD, 0.9% saline); PC: positive group (HFD, 0.5 mg/mL xuezhikang capsules); LD: RFV low dose group (HFD, 10% RFV); MD: RFV medium dose group (HFD, 30% of RFV); HD: RFV high dose group (HFD, 50% of RFV).

**Figure 5 foods-11-04108-f005:**
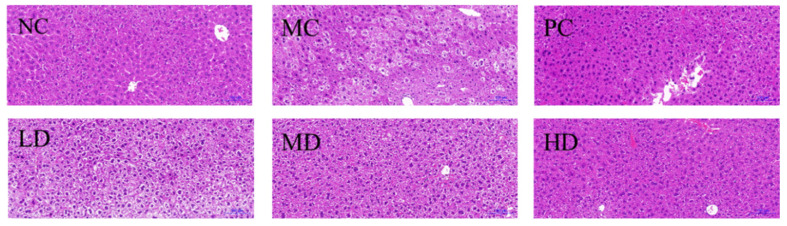
Hepatic lipid accumulation in mice fed experimental diets for 30 days. H&E staining of liver specimens. NC: normal group (normal feed, 0.9% saline); MC: model group (HFD, 0.9% saline); PC: positive group (HFD, 0.5 mg/mL xuezhikang capsules); LD: RFV low dose group (HFD, 10% RFV); MD: RFV medium dose group (HFD, 30% of RFV); HD: RFV high dose group (HFD, 50% of RFV).

**Figure 6 foods-11-04108-f006:**
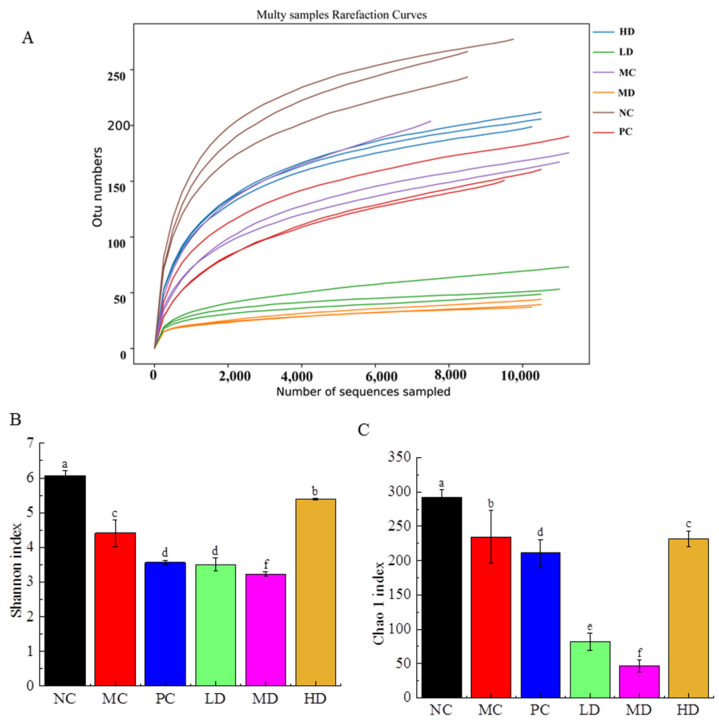
Rarefaction curve, Shannon and Chao 1 index of sequencing reads of the bacterial 16S rDNA gene from the cecum contents samples. (**A**) OUT numbers, (**B**) Shannon indexes, (**C**) Chao 1 indexes. Values are displayed as a bar chart plot with means (expressed as ‘+’), n = 3. a–f: bars with different letters are significantly different at *p* < 0.05 by Tukey’s post hoc test. NC: normal group (normal feed, 0.9% saline); MC: model group (HFD, 0.9% saline); PC: positive group (HFD, 0.5 mg/mL xuezhikang capsules); LD: RFV low dose group (HFD, 10% RFV); MD: RFV medium dose group (HFD, 30% of RFV); HD: RFV high dose group (HFD, 50% of RFV).

**Figure 7 foods-11-04108-f007:**
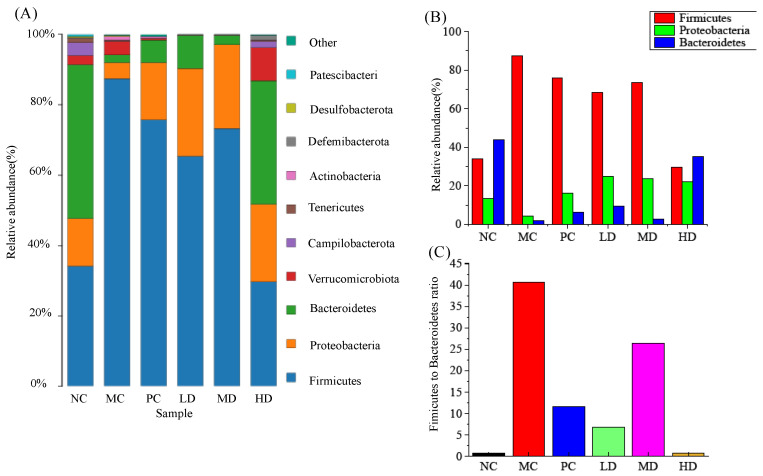
Effects of RFV on the gut microbiota. (**A**) Relative abundance of gut microbiota at the phylum level; (**B**) Relative abundance of *Firmicutes*, *Bacteroidetes* and *Proteobacteria*; (**C**) *Firmicutes* to *Bacteroidetes* ratio. Each phylum is represented by a unique color. NC: normal group (normal feed, 0.9% saline); MC: model group (HFD, 0.9% saline); PC: positive group (HFD, 0.5 mg/mL xuezhikang capsules); LD: RFV low dose group (HFD, 10% RFV); MD: RFV medium dose group (HFD, 30% of RFV); HD: RFV high dose group (HFD, 50% of RFV).

**Figure 8 foods-11-04108-f008:**
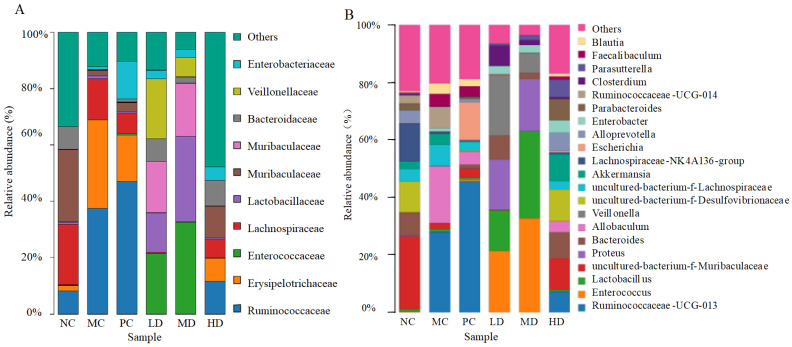
Effects of RFV on gut microbiota of HFD-induced obese mice. Relative abundance of gut microbiota at the family and genus level. (**A**) family level; (**B**) genus level. Each family or genus is represented by a unique color. NC: normal group (normal feed, 0.9% saline); MC: model group (HFD, 0.9% saline); PC: positive group (HFD, 0.5 mg/mL xuezhikang capsules); LD: RFV low dose group (HFD, 10% RFV); MD: RFV medium dose group (HFD, 30% of RFV); HD: RFV high dose group (HFD, 50% of RFV).

**Figure 9 foods-11-04108-f009:**
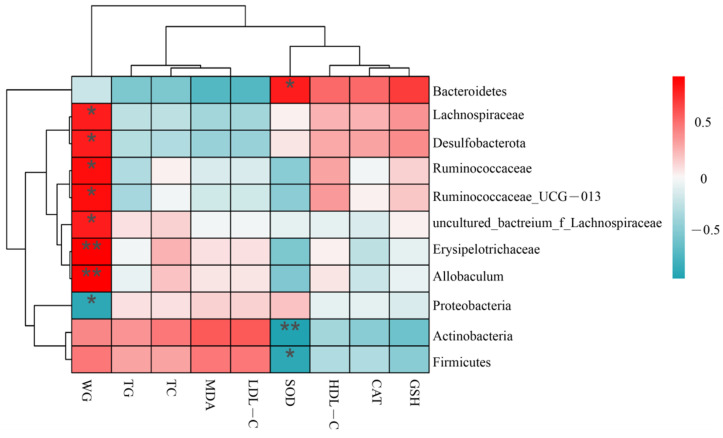
Spearman correlations between obesity-related parameters and microbiota. A blue designates a negative correlation, while a red shows a positive correlation. * (*p* < 0.05), ** (*p* < 0.01). WG in the graph represents weight body gain. WG: weight gain in mice; TC: total cholesterol; TG: triacylglycerol, HDL-C: high-density lipoprotein cholesterol, LDL-C: low-density lipoprotein cholesterol; SOD: total superoxide dismutase; MDA: malondialdehyde; GSH: Glutathione, CAT: catalase.

**Table 1 foods-11-04108-t001:** Nutritional composition of DRRTJ and RFV.

Items	Value of DRRTJ	Value of RFV
Acetic acid	-	35.522 ± 1.125 mg/mL
Lactic acid	1.349 ± 0.011 mg/mL	4.125 ± 0.096 mg/mL
Propionic acid	-	1.753 ± 0.012 mg/mL
Malic acid	8.828 ± 1.230 mg/mL	0.724 ± 0.013 mg/mL
Succinic acid	1.242 ± 0.031 mg/mL	0.465 ± 0.008 mg/mL
Citric acid	0.767 ± 0.009 mg/mL	0.346 ± 0.011 mg/mL
Tartaric acid	0.641 ± 0.010 mg/mL	0.116 ± 0.010 mg/mL
Oxalic acid	0.017 ± 0.037 mg/mL	0.037 ± 0.003 mg/mL
Fumaric acid	-	0.002 ± 0.001 mg/mL
SOD	1279.927 ± 15.341 U/mL	787.53 ± 22.49 U/mL
VC	418.455 ± 13.636 mg/100 mL	367.75 ± 1.58 mg/100 mL
Total phenols	391.511 ± 6.301 mg/100 mL	251.32 ± 1.82 mg/100 mL
Total flavonoids	129.76 ± 3.278 mg/100 mL	98.89 ± 1.88 mg/100 mL

DRRTJ: RRT juice diluted 5 times before fermentation. RFV: *Rosa roxburghii* Tratt fruit vinegar. Values are expressed as the mean ± standard error (n = 3). “-” indicates that it does not exist.

**Table 2 foods-11-04108-t002:** Organ index in mice fed experimental diets for 30 days.

Group	Cardiac Index	Liver Index	Kidney Index	Spleen Index
NC	0.71 ± 0.05 ^b^	4.42 ± 0.26 ^ab^	0.27 ± 0.06 ^b^	1.66 ± 0.16 ^a^
MC	0.60 ± 0.08 ^d^	4.93 ± 0.17 ^a^	0.30 ± 0.05 ^b^	1.40 ± 0.08 ^bc^
PC	0.79 ± 0.06 ^a^	4.35 ± 0.47 ^ab^	0.28± 0.06 ^b^	1.54 ± 0.14 ^ab^
LD	0.69 ± 0.05 ^b^	4.34 ± 0.72 ^ab^	0.27± 0.08 ^b^	1.40 ± 0.16 ^bc^
MD	0.65 ± 0.08 ^c^	4.34 ± 0.89 ^ab^	0.39 ± 0.07 ^a^	1.40 ± 0.16 ^bc^
HD	0.71 ± 0.05 ^b^	4.21 ± 0.63 ^b^	0.32 ± 0.03 ^ab^	1.31 ± 0.17 ^c^

Values are expressed as the mean ± standard error (n = 3). a–d: different letters in the same column indicate significant differences (*p* < 0.05).

## Data Availability

Data are contained within the article.

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
