# Peer review of "Effect and Correlation of Rosa roxburghii Tratt Fruit Vinegar on Obesity, Dyslipidemia and Intestinal Microbiota Disorder in High-Fat Diet Mice"

_foods, 2022, doi:10.3390/foods11244108_

Round 1

Reviewer 1 Report

The manuscript was well prepared and the results in it were novel and very informative to relevant readers. Nevertheless I would like to suggest several things to improve the quality of the manuscript. Please find an attched file and revise according to comments or suggestions if possible.  

Reviewer 2 Report

Thank you for submitting your manuscript " Effect and correlation of Roosa Roxburghii Tratt fruit vinegar  on obesity, dyslipidemia and intestinal microbiota disorder in  high-fat diet mice"  to {Foods}.Following careful assessment of your submission, I enjoy reading it. I found the manuscript of high scientific merit demonstrating applicable work.Preparation of RFV was clear and reproducible. Performing the biochemical, histopathological and gut microbiota is one of the excellent advantage of this study. The data are nicely presented and easily could be followed by readers.

 Minor revision:

Ingredients of the HFD should be mentioned in the Material and Methods section

Author Response

Piease see the attachment
